# Statistical Optimisation of Chemical Stability of Hybrid Microwave-Sintered Alumina Ceramics in Nitric Acid

**DOI:** 10.3390/ma15248823

**Published:** 2022-12-10

**Authors:** Lidija Ćurković, Ivana Ropuš, Hrvoje Cajner, Sanda Rončević, Ivana Gabelica

**Affiliations:** 1Department of Materials, Faculty of Mechanical Engineering and Naval Architecture, University of Zagreb, Ivana Lučića 5, HR-10000 Zagreb, Croatia; 2Energoatest Zaštita Ltd., Potočnjakova 4, HR-10000 Zagreb, Croatia; 3Department of Industrial Engineering, Faculty of Mechanical Engineering and Naval Architecture, University of Zagreb, Ivana Lučića 5, HR-10000 Zagreb, Croatia; 4Department of Chemistry, Faculty of Science, University of Zagreb, Horvatovac 102a, HR-10000 Zagreb, Croatia

**Keywords:** corrosion resistance, hybrid microwave-sintered alumina, response surface methodology, Box–Behnken design

## Abstract

The goal of this research is the statistical optimisation of the chemical stability of hybrid microwave-sintered alumina ceramics in nitric acid. The chemical stability of ceramic materials in corrosive media depends on many parameters, such as the chemical and phase composition of the ceramics, the properties of the aggressive medium (concentration, temperature, and pressure), and the exposure time. Therefore, the chemical stability of alumina ceramics in different aqueous nitric acid solution concentrations (0.50 mol dm^−3^, 1.25 mol dm^−3^, and 2.00 mol dm^−3^), different exposure times (up to 10 days), as well as different temperatures (25, 40, and 55 °C), was investigated, modelled, and optimised. The chemical stability of high purity alumina ceramics (99.8345 wt.% of Al_2_O_3_) was determined by measuring the amount of eluted ions (Al^3+^, Ca^2+^, Fe^3+^, Mg^2+^, Na^+^, and Si^4+^) obtained by inductively coupled plasma atomic emission spectrometry. The changes in the density of alumina ceramics during the chemical stability monitoring were also determined. The Box–Behnken approach was employed to reach the optimum conditions for obtaining the highest possible chemical stability of alumina at a given temperature range, exposure time, and molar concentration of nitric acid. It was found that an increase in exposure time, temperature, and nitric acid concentration led to an increase in the elution of ions from hybrid microwave-sintered alumina. Higher amounts of eluted ions, Al^3+^ (14.805 µg cm^−2^), Ca^2+^ (7.079 µg cm^−2^), Fe^3+^ (0.361 µg cm^−2^), Mg^2+^ (3.654 µg cm^−2^), and Na^+^ ions (13.261 µg cm^−2^), were obtained at 55 °C in the 2 mol dm^− 3^ nitric acid. The amount of eluted Si^4+^ ions is below the detection limit of inductively coupled plasma atomic emission spectrometry. The change in the alumina ceramic density during the corrosion test was negligible.

## 1. Introduction

Alumina (Al_2_O_3_) is an example of oxide ceramics widely used due to its excellent combination of mechanical, wear, electrical, thermal, and chemical properties [1,2]. These properties make them useful materials in high-tech industries, in devices operating under extreme conditions (high temperature, high pressure, and corrosive media), in aerospace engineering, in safety systems, and others [3,4].

The final properties of ceramic materials are determined in the manufacturing process, starting from powder shaping to sintering [5]. Sintering can be carried out via conventional methods (in an electric kiln) or non-conventional ones, such as microwave sintering. Microwave heating is based on the oscillatory motion of charged particles driven by an electromagnetic field at the frequency of the microwave source [6]. A susceptor (high dielectric loss factor material) is often used to potentiate the heating of the green body to the temperature of significant coupling with microwaves. The combination of conventional heating and microwave heating is referred to as hybrid microwave heating [7]. This non-conventional sintering method saves time and energy due to its very high heating rates. Sintered samples have a finer microstructure which contributes to improved mechanical properties [8], such as a higher Vickers hardness and fracture toughness, and lower brittleness [9].

During sintering, two mechanisms occur: densification and coarsening of the grains in the ceramic material. The space between the grains is called the grain boundary, which can be an amorphous and/or crystalline phase [10,11]. These grain boundaries are the primarily attacked areas in aggressive, corrosive liquid media due to the dissolution of cations previously segregated at the grain boundaries. Consequently, corrosion can cause leaching of the grain boundaries and the prolapse of grains from the ceramic surface [12]. Moreover, it may lead to the deterioration of mechanical and electrical properties [13,14].

Corrosion in metals is an electrochemical process; the solubility of the material is the determining factor in the level of ceramic corrosion. Chemical composition and microstructure are the key factors for low corrosion rates. It is important to point out that no standardised test procedures are known for the determination of ceramic corrosion in the liquid phase [15]. Therefore, various studies have explored the factors that influence the corrosion resistance of ceramics. Schacht et al. [16] reported that the dominant corrosion mechanisms are intergranular attack and the dissolution of aluminium oxide in aqueous acidic solutions (HCl, H_2_SO_4_, H_3_PO_4_) under hydrothermal conditions. The authors concluded that the intensity of the dissolution of grain boundaries depends on the purity of the ceramics since impurities move to grain boundaries due to their low solubility in aluminium oxide. The corrosion behaviour of silicon nitride materials in an aqueous solution of acid (H_2_SO_4_) was investigated by Schlim et al. [17]. The authors noticed that the corrosion resistance of silicon nitride materials in acids is highest for materials with grain boundary phases consisting of strongly linked networks. As a result, they came to the conclusion that corrosion resistance was directly related to the composition and structure of the grain boundary phase. Wang and Yin [18] investigated the corrosion behaviour of the porous alumina-based ceramic core materials in potassium hydroxide and sodium hydroxide solutions. More severe corrosion was observed in porous ceramics than in dense ceramics because porous ceramics have a larger specific surface area and, thus, a larger area is in contact with the alkali medium, with temperature as a critical factor in the corrosion process. Alumina corrosion resistance to acidic and base solutions is also dealt with in the literature [11,16,19,20]. The chemical stability of ceramics may be determined by monitoring the morphology [21,22], weight loss or gain [20,22], and mechanical properties [12,21] of ceramics after exposure to an aggressive environment. In addition, the amount of ions eluted from the investigated ceramic material in the corrosive medium can indicate its chemical stability [23].

A growing number of possibilities for the application of advanced ceramics demands a wider knowledge of corrosion processes and the ability to control them. Therefore, it is necessary to optimise the process of predicting the corrosion resistance of ceramics. The corrosion resistance of alumina sintered in electric kilns was previously tested by varying one parameter at a time, which is time consuming and costly [23]. This approach does not include interactions between the independent parameters of the process. However, the use of the response surface methodology (RSM) approach, which is a multidimensional system, allows us to examine and optimise the interactions between the factors of the process by implementing a statistical technique, as shown in our previous paper [24].

The response surface methodology is useful for the optimisation, planning, development, and improvement of processes where one response, or more of them, depends on multiple variables. It considers different designs for screening and optimisation purposes [25,26,27,28]. Some of the most common forms of the response surface method are central composite design (CCD), face-centred composite design (FCC), Box–Behnken design (BBD), and three-level factorial design [25,29]. The limitation of the response surface method lies in the fact that the obtained response surface is valid only for the observed ranges of the selected factors.

The Box–Behnken design considers all factors situated on a hypersphere (equally distant from the central point) simultaneously and not under extreme conditions (vertices of the cube) [30]. The advantage of the Box–Behnken design is the possibility of its use in cases of limitations of the experimental area and a reduced number of levels of each factor, which reduces the number of experiment conditions (recommended for models with up to four factors). The number of experiments required to implement the Box–Behnken experiment plan may be expressed by the following equation [31]:(1)$$N=k^{2}+k+n_{c}$$
where *N* is the number of experiments, *k* is the number of factors (independent variables), and *n*_c_ is the number of repeated experiments at the central point.

The impact of nonconventional sintering methods (e.g., hybrid microwave sintering) on the corrosion resistance of ceramics has not been investigated in a broader sense until now, so this research tends to contribute to its understanding. Methods for determining the liquid-phase corrosion of ceramics include the following conditions: (i) the storage of test samples in the corrosive media at room temperature (over a test period of between 10 and 100 days); (ii) storage of the test samples in the reflux cooler at higher temperatures (over a test period of between 5 and 50 days); and (iii) storage in autoclaves at higher temperatures and pressure (test period 1 to 5 days) [15]. To reach that goal, a static corrosion test was conducted at different temperatures and concentrations of nitric acid for 10 days according to the Box–Behnken design. The measured bulk density of alumina ceramics and the amounts of Al^3+^, Ca^2+^, Fe^3+^, Mg^2+^, Na^+^, and Si^4+^ ions eluted in nitric acid from alumina were statistically analysed, and optimum conditions were suggested to achieve maximal density and the lowest amounts of eluted ions. Additionally, the obtained results were compared with previous results on corrosion of the same alumina ceramics sintered in an electric kiln (conventional method) under the same corrosion test conditions.

## 2. Materials and Methods

### 2.1. Preparation of Alumina Ceramics

In this research, a commercial alumina raw powder (Alteo, France) was used as the starting material for the production of samples. According to the manufacturer’s specification, the as-received powder has an average particle size of 0.40 μm and a surface area of 8.0 m^2^ g^−1^. Chemical composition was determined by inductively coupled plasma (ICP) spectrometry according to the manufacturer’s declaration; it is shown in Table 1.

Chemical composition indicates that the investigated alumina ceramics has high purity, with 99.8345 wt.% of aluminium oxide; the rest, i.e., 0.1655 wt.%, contains magnesium oxide as a sintering aid and the usual impurities, i.e., ferric oxide, calcium oxide, silicon dioxide, and sodium oxide. Magnesium oxide used as a sintering aid prevents abnormal crystal grain growth.

Alumina granules were formed by the spray drying process. The particle size distribution of the spray-dried alumina granules was determined by sieve analysis according to the ISO 3310-1 standard. The obtained results are shown in Figure 1.

The particle size distribution follows a normal distribution (Figure 1), showing that 96% of the particles are in the range of 50 to 300 µm. The particle shape and size are important properties of the powder because they ensure the proper filling of the mould in the process of green body production [32].

The granules were then isostatically cold pressed into cylindrical pellets with diameters of 10 mm and heights of 20 mm. Both production processes were carried out at Applied Ceramics Ltd., Sisak, Croatia. A hybrid microwave furnace (OVER—Industrial Electronics and Trade, Kerestinec, Croatia) with a magnetron of 1.5 kW and 2.45 GHz frequency was used for the green pellet sintering. Sintering of alumina ceramic samples in a hybrid microwave oven was carried out according to the following regime: samples were heated (i) from room temperature to 1300 °C at a rate of 27 °C min^−1^, (ii) from 1300 °C to 1600 °C at a rate of 4 °C min^−1^, followed by (iii) isothermal holding at a temperature of 1600 °C for 1 h, and (iv) natural cooling of samples in the furnace to room temperature according to the speed determined by the characteristics of the furnace. After sintering, the dimensions (diameter and height) of all alumina samples were measured, and the surface area was calculated for each sample.

### 2.2. Characterisation of Alumina Ceramics

Powder X-ray diffraction (XRD) was performed using a Shimadzu XRD6000 diffractometer with CuKα radiation (Shimadzu Corporation, Kyoto, Japan) to determine the phase composition of aluminium oxide granules under an accelerating voltage of 40 kV and a current of 30 mA with a step size of 0.02° between 10° and 80° (2θ) and a counting time of 0.6 s.

Sintered samples were prepared according to the standard ceramographic technique [23] and then coated with a layer of gold; subsequently, the analysis of morphology was conducted using a scanning electron microscope (SEM) (Tescan Vega TS5136LS, Czech Republic).

The Archimedes method was used according to ASTM C373-88 to determine the bulk density of sintered alumina (Mettler Toledo GmbH, Switzerland, density kit MS-DNY-43).

An hardness tester Wilson Wolpert Tukon 2100B (Instron, Grove City, PA, USA) was used to determine the Vickers hardness of the sintered samples. Vickers hardness measurements (*HV*1) were performed using indentation loads of 9.807 N for 15 s according to the ASTM E92-72 standard. Diagonals were measured using an optical microscope Olympus BH (Olympus Imaging Corp., Japan) immediately after unloading. Measurements were conducted at room temperature 10 times per sample. The crack dimensions were less than one-tenth of the thickness of the samples [33].

Fracture toughness was determined by calculating the ratio of the Vickers crack length and half of the Vickers indentation diagonal (*c*/*a*), which shows the crack type as an indirect indicator of the toughness of ceramics [34,35,36].

### 2.3. Monitoring of Alumina Corrosion in an Aqueous Solution of Nitric Acid

Sintered alumina pellets were rinsed with alcohol and dried in a sterilizer at 150 ± 5 °C for 4 h. Marked polypropylene (PP) tubes were filled with 10 cm^3^ of nitric acid at the appropriate concentration (0.50, 1.25, and 2.00 mol dm^−3^). The samples were then immersed in the acid solutions, and the PP tubes were sealed. The corrosion test was carried out at 25, 40, and 55 °C and different exposure times (up to 10 days).

Afterwards, alumina samples were removed from the tubes, rinsed with distilled water and dried in an oven for three hours at 150 °C. Finally, the bulk density of the alumina samples after the corrosion test was measured.

Mechanisms responsible for the corrosion processes were subsequently observed by determining the concentration of eluted ions in the corrosive aqueous solution of nitric acid. The concentration of eluted ions (Al^3+^, Ca^2+^, Fe^3+^, Mg^2+^, and Na^+^) was determined by inductively coupled plasma atomic emission spectroscopy (ICP—AES) (Teledyne Leeman Labs, Hudson, NH, SAD), while the Si^4+^ cations were under the quantification limit [LOQ (Si^4+^) < 0.45 µg g^−1^] [37]. The obtained results were expressed as the amount of eluted ions (M^n+^: Al^3+^, Ca^2+^, Fe^3+^, Mg^2+^, and Na^+^) in µg per square centimetre of the tested alumina surface area (µg M^n+^ cm^−2^).

### 2.4. Design of Experiments of Monitored Alumina Corrosion Resistance

The Box–Behnken design was used for modelling and optimisation purposes. Its economically beneficial site, requiring a minimum of only three levels for each factor, i.e., lower value (−1), central value (0), and higher value (+1) [25,38], as well as the resources and time limitations of the present experiment, made the Box–Behnken design a preferable design for this research.

Previous research and literature findings [12,19,23,39,40] suggest several parameters influencing the corrosion of ceramic materials: time, temperature, properties of media (e.g., pH, static, flowing) and properties of the observed ceramic material (e.g., chemical composition and microstructure). In this research, three parameters at three levels were observed: time, temperature, and concentration of the acid media, with five runs at the **centre** point. The factors and design points were chosen according to the Box–Behnken design and are listed in Table 2 and Table 3.

Design Expert^®^ software (version 13) by Stat-Ease Inc. (Minneapolis, USA) was used to facilitate the design and modelling by the Box–Behnken design. However, one must remember that a statistical interpretation has to be based on the subject knowledge applied within the experimentally predetermined boundaries (Table 2). Therefore, six responses were recorded from the experiment: amounts of Al^3+^, Mg^2+^, Ca^2+^, Na^+^, and Fe^3+^ ions eluted from the alumina and the density of alumina.

## 3. Results and Discussion

### 3.1. Properties of Alumina Ceramics

Characteristic peaks in the X-ray diffractogram (Figure 2) show that alumina granules consist of only one crystalline phase, which is α-Al_2_O_3_ (corundum) (ICDD PDF#46-1212).

The hybrid microwave sintering process led to the formation of fine grains and small pores in the bulk alumina material (Figure 3). The obtained grains do not show a specific orientation or a uniform grain size. The calculated average grain size of 2.1 µm, determined by the line intercept method [41,42], is in accordance with findings in the literature [43,44,45,46]. The average grain size of the same alumina ceramics sintered in a high-temperature electric kiln (conventional sintering) at the same temperature was greater, i.e., 7.6 μm [24]. The grain size decreases in hybrid microwave sintering compared to conventional sintering due to the shorter sintering time. The measured bulk density was 3.784 ± 0.028 g cm^−3^ (Table 4), while the relative porosity was 5.1 ± 0.5%.

The Vickers hardness was calculated according to the following equation (Figure 4) [47]:(2)$$HV=\alpha \frac{F}{d^{2}}$$
where *HV* is the Vickers hardness, α is the geometrical constant of the indenter, i.e., 0.1891 for the Vickers diamond pyramid, *F* is the applied load (N), and *d* is the mean value of the indentation diagonals (mm). The Vickers hardness was measured under a load of 9.807 N, which caused the development of surface cracks. The obtained ratio of *c*/*a* is 1.96 ± 0.26, which is less than 2.5 (Table 4); this indicates the Palmqvist crack system that was used to determine fracture toughness according to Casellas (Equation (3)), Niihara et al. (Equation (4)), and Shetty et al. (Equation (5)) [47]:(3)$$K_{\mathrm{Ic}}=0.024\cdot \frac{F}{c^{1.5}}\cdot {\left(\frac{E}{HV}\right)}^{0.5}$$
(4)$$K_{\mathrm{Ic}}=0.0089\cdot {\left(\frac{E}{HV}\right)}^{0.4}\cdot \frac{F}{a\cdot l^{0.5}}$$
(5)$$K_{\mathrm{Ic}}=0.0319\cdot \frac{F}{a\cdot l^{0.5}}$$
where *F* is the load applied during the Vickers test (N); *c* is the crack length from the centre of the indentation to the crack tip (m); *E* is Young’s modulus (GPa); *HV* is the Vickers hardness (GPa); and *l* is the crack length measured from the vertices of the indentation to the crack tip (m). Figure 4 shows the characteristic values used to calculate the Vickers fracture toughness (*K*_IC_) of brittle materials.

**Table 4 materials-15-08823-t004:** Properties of sintered Al_2_O_3_ samples: density, hardness (*HV*1), ratio of c/a, and Vickers indentation fracture toughness.

Sample	*ρ*, g cm^−3^	*HV*1	*c/a*	*K*_IC_, MPa m^1/2^		
				Casellas [48]	Niihara et al. [48]	Shetty et al. [48]
Al_2_O_3_	3.784 ± 0.028	1815 ± 51	1.96 ± 0.26	6.45 ± 1.43	4.91 ± 0.81	5.15 ± 0.89

The Vickers hardness (*HV*1), the *ratio of c/a*, and the Vickers indentation fracture toughness (*K*_IC_) values are listed in Table 4. In comparison to the hardness and indentation fracture toughness of the same alumina ceramics sintered in an electric kiln, which amount to 1762 ± 77 and 5.44 ± 0.93 MPa m^1/2^ (Casellas), respectively, the values for the ceramic samples sintered in a hybrid microwave oven are higher (Table 4) [24].

### 3.2. Modelling of the Amount of Eluted Ions and Alumina Density

A dataset with design values, which is obtained from the conducted experiments, is shown in Table 5.

A static corrosion test of non-conventionally sintered alumina samples was conducted to determine the corrosion resistance of alumina to the action of nitric acid in three different concentrations for 10 days at different temperatures. The obtained results are expressed as the amount of eluted ions (M^n+^: Al^3+^, Ca^2+^, Fe^3+^, Mg^2+^, and Na^+^) in µg per square centimetre of the tested alumina surface area (µg M^n+^ cm^−2^).

Analysis of variance (ANOVA), a numerical method for model validation, was used to validate the adequacy of the regression models and the significance of the estimated coefficients. In the case that the *p*-value is less than 0.05, the model is considered to be significant (Table 6). In such a case, the model succeeded in explaining more variability than it left unexplained.

A polynomial regression model was used to analyse the relationship between de-pendent and independent variables, while ANOVA was used for the amounts of Al^3+^, Mg^2+^, Ca^2+^, Na^+^, and Fe^3+^ eluted ions, as well as the density of alumina ceramics. The ANOVA table (Table 6) shows the response regarding the amounts of eluted Al^3+^ ions.

Different response transformations were applied in order to successfully stabilise the variances. The natural log for Al^3+^, the base 10 log for Ca^2+^, and the inverse square root for Fe^3+^, Mg^2+^, and Na^+^ were applied. On the other hand, the density response needed no transformation.

Regression equations for each response (amount of eluted ions and density) are given in Table 7, while the normal plot of the residuals for the two responses can be seen in Figure 5. Response surface plots, as graphic representations of the regression models, are shown in Figure 6.

The models of the amount of eluted ions may be considered adequate because they have a suitable (normal) distribution of residuals. As an example, the distribution of residuals for the model of eluted aluminium ions is shown in Figure 5A. Furthermore, the models exhibit a high signal-to-noise ratio (>87) and a high coefficient of determination (*R*^2^ ≥ 99.8%).

The model of density has an acceptable shape for the residuals as a function of the predicted values (Figure 5B) and signal-to-noise ratio (6.7 > 4). The *R*^2^ value of 80% may be considered high enough for obtaining a precise prediction of the density value with the obtained model.

However, high *R*^2^ values and a normal distribution of the residuals of all responses demonstrate the adequacy of the obtained models and prediction capability. Furthermore, Figure 6 depicts the response surface plots of the regression models in a 3D diagram with time and concentration as constant variables at a temperature of 40 °C.

### 3.3. Optimisation and Verification of Alumina Ceramics Corrosion Resistance in Nitric Acid

Numerical and graphic optimisation of the observed corrosion resistance process was made by the multi-criteria methodology named the Derringer function or desirability function [48]. The input data of each individual response (*y_i_*) were transformed into an individual dimensionless desirability function, *d_i_* (desirability scale), ranging from zero (non-desirable) to 1 (highly desired responses), after which any improvement would be insignificant. Taking into account the desirability functions of individual responses (*d_i_*), it is possible to obtain the overall desirability (*D*) according to the equation below:(6)$$D=\sqrt[{m}]{\left(d_{1}\cdot d_{2}\cdot \ldots \cdot d_{m}\right)}$$
where D is the overall desirability, d_m_ is the individual desirability function of the m response, and m is the number of responses [29].

The obtained regression models (Table 7) underwent verification to check their adequacy by comparing predicted values with experimental values [49,50]. All results fall within the 95% confidence interval of the mean, which proves that they satisfy the 95% prediction interval. Consequently, the models are applicable and useful for predicting the response’s values.

The increase in time and temperature led to an increase in the elution of all ions (Figure 6A–E). Furthermore, the increase in the nitric acid concentration influenced the increase in the amounts of all eluted ions. The maximum values of eluted Al^3+^, Ca^2+^, Na^+^, Fe^3+^, and Mg^2+^ ions in nitric acid are reached at the highest temperature and longest immersion time in the highest concentration of nitric acid in the following order: Fe^3+^ < Mg^2+^ < Ca^2+^ < Na^+^ < Al^3+^. The change in the alumina ceramic density during the corrosion test was negligible (Figure 6F). For the same alumina samples sintered in an electric kiln under the conditions of the corrosion test, it was found that the amounts of eluted ions from the alumina ceramics are in the following order: Fe*^3+^* < Mg*^2+^* < Na*^+^* < Al*^3+^* < Ca*^2+^* [24].

The alumina raw powder investigated in this research consisted of 99.8345 wt.% of aluminium oxide and 0.1655 wt.% of impurities (calcium oxide, ferric oxide, sodium oxide, and silicon dioxide) and a sintering aid (magnesium oxide). Magnesium oxide prevents abnormal crystal grain growth. During sintering, impurities may be incorporated into a highly packed alumina hexagonal crystal lattice (Figure 7) due to the one-third octahedral vacancy that is not filled by aluminium cations (Al^3+^). Impurities may also move to the grain boundaries because their amount in the investigated ceramic is not sufficient to create another crystal phase at the grain boundaries [10]. Therefore, the corrosion of alumina ceramics is in keeping with its microstructure and the distribution of calcium oxide, ferric oxide, magnesium oxide, sodium oxide, and silicon dioxide in the alumina. The difference between the ionic radius of Ca^2+^, Fe^3+^, Mg^2+^, Na^+^, and Si^4+^ cations and the Al^3+^ cation (Table 8) defines their solubility in the alumina [51]. Since the impurities in alumina ceramics (calcium oxide, ferric oxide, sodium oxide, and silicon dioxide) and the sintering aid (magnesium oxide) have a low solubility in the alumina, they move to the grain boundaries during the sintering process [16,23].

Numerical and graphic optimisation showed the highest corrosion resistance of the investigated alumina at the beginning of the corrosion test, i.e., after 24 h of its exposure to 0.50 mol dm*^−^*^3^ nitric acid at 25 °C, with a desirability of 96%. The same result was obtained for the same alumina ceramics sintered in an electric furnace under the same corrosion conditions [24].

The desirability function (Figure 8) shows that alumina ceramics are more corrosion resistant at lower temperatures and concentrations, after a shorter time of exposure to nitric acid, while higher concentrations and temperatures of nitric acid and a longer exposure time have the opposite effect on the elution of ions from hybrid microwave-sintered alumina.

Experimental values are in good agreement with the predicted values for all models (Table 9), meaning that all models may be considered adequate for the prediction of the optimum corrosion resistance of alumina ceramics.

A comparison of the verification of the experiment results with the predicted values is shown in Table 10. Some deviations in the results are evident, but they can still be considered adequate while optimum parameters are placed outside of the Box–Behnken design points. The 96% desirability function value means that 96% of the maximum response value is achieved considering the given constraints and criteria. An alternative optimum may be defined after 24 h of alumina exposure to 0.50 mol dm^−3^ nitric acid at 40 °C with a lower, but still acceptable desirability (88%). The verification of the experimental and calculated values of eluted ion amounts and alumina ceramic density at assessed optimal corrosion parameters can also be seen in Table 10, where predicted and experimental results are within or near 95% of the confidence interval. The single point prediction interval is much wider than the CI, so it can be concluded that the verification points are in the range; this supports the hypothesis that the design is suitable.

## 4. Conclusions

The Box–Behnken design was applied to conduct an experiment on the chemical stability of alumina ceramics sintered by means of a hybrid microwave sintering process.The corrosion resistance of alumina ceramics to 0.5, 1.25, and 2.00 mol dm^−3^ nitric acid at 25, 40, and 55 °C for up to 10 days was investigated.The inverse behaviour of the density values in relation to the amounts of eluted ions was demonstrated by regression models accompanied by surface plots.The optimal corrosion resistance parameters obtained for the investigated alumina are the minimum exposure time (24 h) of alumina to the 0.50 mol dm^−3^ nitric acid at 25 °C with a desirability of 96%. After 24 h of alumina exposure to the 0.50 mol dm^−3^ nitric acid at 40 °C, a second optimum, with a lower, but still acceptable desirability (88%) occurred.The amounts of eluted ions from the alumina ceramics are in the following order: Fe^3+^ < Mg^2+^ < Ca^2+^ < Na^+^ < Al^3+^.The highest amounts of eluted ions, Al^3+^ (14.805 µg cm^−2^), Ca^2+^ (7.079 µg cm^−2^), Fe^3+^ (0.361 µg cm^− 2^), Mg^2+^ (3.654 µg cm^−2^), and Na^+^ (13.261 µg cm^−2^), were obtained at 55 °C in 2 mol dm^−3^ nitric acid. The amount of eluted Si^4+^ ions is below the detection limit of ICP-AES.The change in alumina ceramic density during the corrosion test was negligible.In general, it can be concluded that the corrosion of alumina ceramics sintered in hybrid microwave kilns shows good chemical stability under test conditions. Corrosion can be mostly attributed to the dissolution of segregated impurities (calcium oxide, ferric oxide, sodium oxide, and silicon dioxide) and the sintering aid (magnesium oxide) from the grain boundaries of the alumina ceramics.This study demonstrated that an unconventional sintering process (hybrid microwave) can produce high-purity alumina ceramics with a favourable microstructure for use in a corrosive nitric acid medium.

## Figures and Tables

**Figure 1 materials-15-08823-f001:**
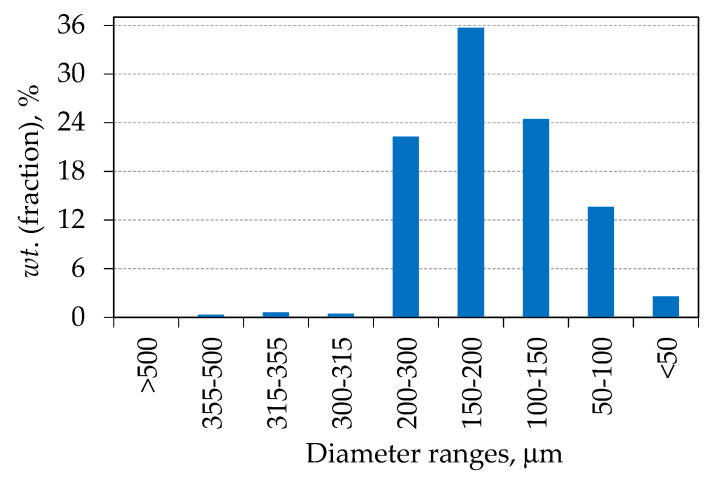
Particle size distribution of the spray-dried alumina granules.

**Figure 2 materials-15-08823-f002:**
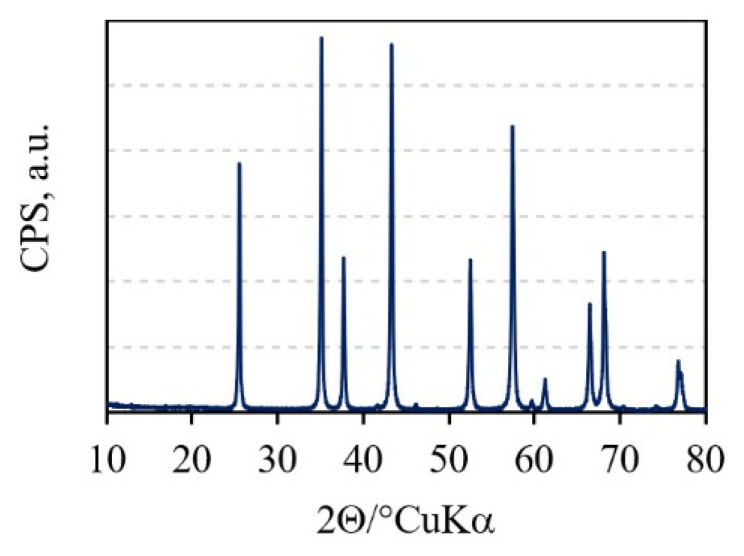
XRD pattern of alumina granules.

**Figure 3 materials-15-08823-f003:**
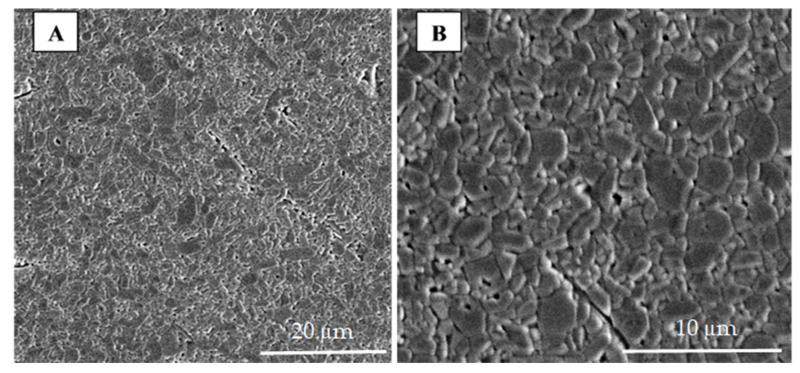
SEM image of the alumina ceramics sintered in a hybrid microwave kiln at different magnifications: (**A**) 2500×, (**B**) 6000×.

**Figure 4 materials-15-08823-f004:**
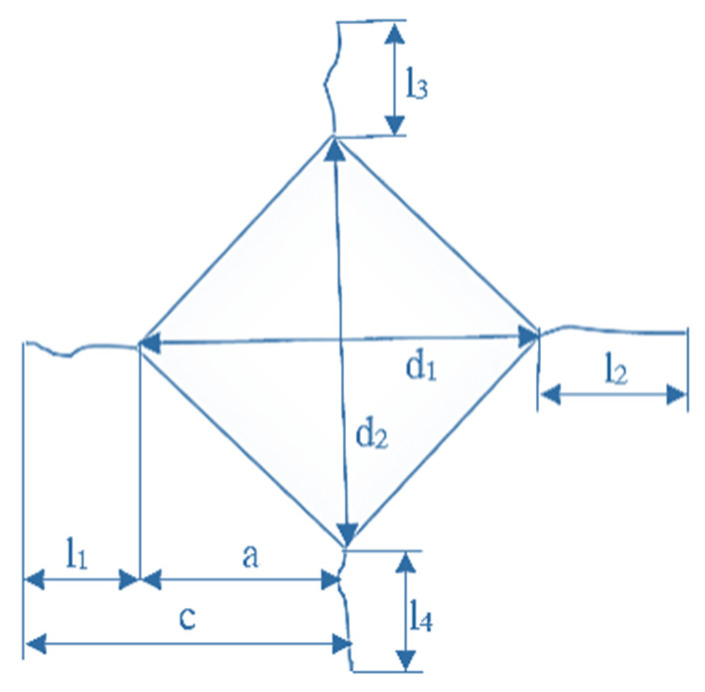
The characteristic values used to calculate the Vickers indentation fracture toughness (*K*_IC_) of brittle materials.

**Figure 5 materials-15-08823-f005:**
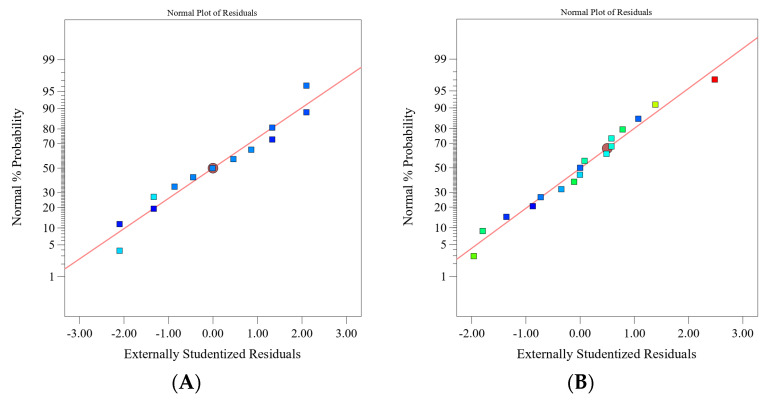
Normal plot of residuals: (**A**) amounts of aluminium ions, and (**B**) density.

**Figure 6 materials-15-08823-f006:**
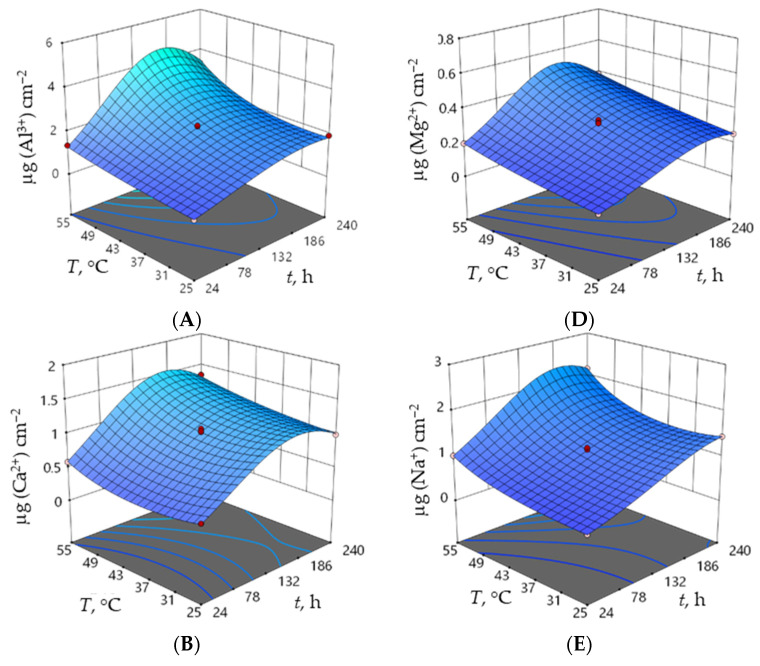
Response surface plots of the regression models of the amounts of eluted ions (**A**) Al^3+^, (**B**) Ca^2+^, (**C**) Fe^3+^, (**D**) Na^+^, (**E**) Mg^2+^ and (**F**) the density of alumina ceramics at a constant concentration (1.25 mol dm^− 3^) of nitric acid.

**Figure 7 materials-15-08823-f007:**
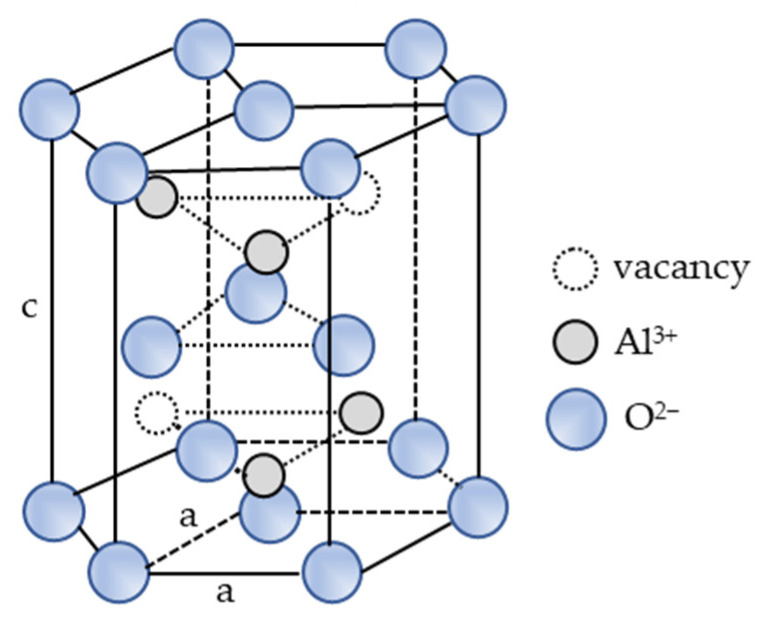
Corundum structure of alpha-alumina (*α*-Al_2_O_3_).

**Figure 8 materials-15-08823-f008:**
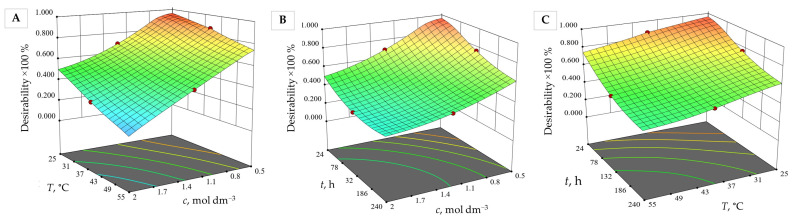
Desirability function of the amount of eluted ions and alumina ceramic density dependent on (**A**) nitric acid concentration and temperature at constant time (132 h), (**B**) time and nitric acid concentration at constant temperature (40 °C), and (**C**) time and temperature at constant nitric acid concentration (1.25 mol dm^−3^).

**Table 1 materials-15-08823-t001:** Chemical composition of the alumina used in this research.

Component	Fe_2_O_3_	CaO	SiO_2_	MgO	Na_2_O	Al_2_O_3_
wt.%	0.018	0.02	0.0325	0.045	0.05	balance

**Table 2 materials-15-08823-t002:** Factors used in the Box–Behnken design.

Independent Variable	−1 Level	0	+1 Level
*c* (HNO_3_), mol dm^−3^	0.50	1.25	2.00
*T*, °C	25	40	55
*t*, h	24	132	240

**Table 3 materials-15-08823-t003:** Experimental plan according to the Box–Behnken design.

No	*c* (HNO_3_), mol dm^−3^	*T*, °C	*t*, h
1	0.50	55	132
2	2.00	55	132
3	1.25	25	240
4	1.25	40	132
5	0.50	40	240
6	1.25	40	132
7	1.25	40	132
8	1.25	40	132
9	1.25	40	132
10	0.50	40	24
11	1.25	25	24
12	2.00	25	132
13	1.25	55	240
14	2.00	40	240
15	0.50	25	132
16	2.00	40	24
17	1.25	55	24

**Table 5 materials-15-08823-t005:** All experimental values used for response surface plots and regression equations.

Run	*c*,	*T*,	*t*,	Al^3+^,	Ca^2+^,	Fe^3+^,	Mg^2+^,	Na^+^,	*ρ*,
	mol dm^−3^	°C	h	µg cm^−2^	µg cm^−2^	µg cm^−2^	µg cm^−2^	µg cm^−2^	g cm^−3^
1	0.50	55	132	0.775	0.690	0.021	0.113	0.910	3.781
2	2.00	55	132	14.805	7.079	0.361	3.654	13.261	3.767
3	1.25	25	240	1.918	0.990	0.029	0.253	1.440	3.783
4	1.25	40	132	2.146	0.982	0.025	0.311	1.079	3.772
5	0.50	40	240	0.763	0.731	0.015	0.100	0.892	3.776
6	1.25	40	132	2.189	1.004	0.024	0.310	1.128	3.779
7	1.25	40	132	2.239	1.040	0.024	0.315	1.135	3.774
8	1.25	40	132	2.294	1.029	0.023	0.335	1.151	3.779
9	1.25	40	132	2.338	1.079	0.023	0.316	1.189	3.769
10	0.50	40	24	0.310	0.101	0.006	0.038	0.250	3.785
11	1.25	25	24	0.645	0.463	0.011	0.095	0.472	3.792
12	2.00	25	132	3.982	1.301	0.041	0.495	1.839	3.796
13	1.25	55	240	3.352	1.340	0.040	0.388	2.154	3.771
14	2.00	40	240	3.633	2.144	0.075	0.841	2.221	3.771
15	0.50	25	132	0.402	0.618	0.008	0.067	0.396	3.809
16	2.00	40	24	1.690	1.688	0.025	0.663	1.415	3.778
17	1.25	55	24	1.353	0.582	0.019	0.195	1.014	3.784

**Table 6 materials-15-08823-t006:** ANOVA for the amount of Al^3+^ ions eluted from alumina ceramics after exposure to nitric acid.

Source	Sum of Squares	df	Mean Square	*F*-Value	*p*-Value(Prob > F)
Model	14.06	10	1.41	477.06	<0.0001 *
A-Concentration	2.65	1	2.65	900.17	<0.0001
B-Temperature	0.97	1	0.97	329.43	<0.0001
C-Time	0.69	1	0.69	235.56	<0.0001
AB	0.11	1	0.11	36.54	0.0009
A^2^	0.18	1	0.18	62.07	0.0002
B^2^	0.07	1	0.07	23.96	0.0027
C^2^	1.07	1	1.07	364.30	<0.0001
AB^2^	0.49	1	0.49	167.27	<0.0001
B^2^C	0.01	1	0.01	4.65	0.0745
BC^2^	0.06	1	0.06	19.14	0.0047
Residual	0.018	6	0.003		
Lack of Fit	0.013	2	0.007	5.41	0.0728 **
Pure Error	0.005	4	0.001		
Cor Total	14.07	16			

* significant; ** not significant; *α* = 0.05.

**Table 7 materials-15-08823-t007:** Regression equations of the amount of eluted ions and the density of alumina ceramics expressed in coded values.

Response	Regression Equations
ln (µg Al^3+^ cm^−2^)	0.81 + 0.81A + 0.49B + 0.42C + 0.16AB − 0.21A^2^ + 0.13B^2^ − 0.51C^2^ + 0.50AB^2^ + 0.08B^2^C − 0.17BC^2^
log (µg Ca^2+^ cm^−2^)	0.011 + 0.42A + 0.06B + 0.24C + 0.17AB − 0.19AC + 0.05A^2^ + 0.09B^2^ − 0.21C^2^ + 0.14A^2^B − 0.09AB^2^ − 0.07B^2^C
(µg Fe^3+^ cm^−2^)^−0.5^	6.49 − 2.89A − 0.81B − 1.54C + 0.31AB + 0.56AC + 0.36BC + 0.32A^2^ − 0.59B^2^ + 1.06C^2^− 1.14A^2^B − 0.38A^2^C
(µg Mg^2+^ cm^−2^)^−0.5^	1.78 − 1.50A − 0.34B − 0.48C + 0.46AC + 0.15BC + 0.40A^2^ + 0.48C^2^ − 0.10A^2^B − 0.05A^2^C + 0.28AB^2^
(µg Na^+^ cm^−2^)^−0.5^	0.94 − 0.40A − 0.15B − 0.23C + 0.19AC + 0.08BC + 0.06A^2^ − 0.09B^2^ + 0.14C^2^ − 0.10A^2^B − 0.05A^2^C
*ρ*	3.78 − 0.01A − 0.01B − 0.01C + 0.01B^2^

A—*c* (HNO_3_), mol dm^−3^; B—*T*, °C; C—*t*, h.

**Table 8 materials-15-08823-t008:** Ionic radius of cations present in Al_2_O_3_ [51].

Cation	Al^3+^	Na^+^	Ca^2+^	Mg^2+^	Fe^3+^	Si^4+^
*r*_ion_, pm	53.5	102	100	72	64.5	40

**Table 9 materials-15-08823-t009:** Verification of experimental and calculated values of the eluted ion amounts and the density of alumina ceramics at randomly chosen parameters of corrosion.

No. of Verifications	Response	Experimental Values	Mean Predicted Value	Low CI (95%)	High CI (95%)
1	Experimental parameters: 0.50 mol dm^−3^ HNO_3_, 25 °C, 132 h				
	µg (Al^3+^) cm^−2^	0.402	0.402	0.352	0.460
	µg (Ca^2+^) cm^−2^	0.618	0.618	0.564	0.678
	µg (Fe^3+^) cm^−2^	0.008	0.008	0.007	0.008
	µg (Mg^2+^) cm^−2^	0.067	0.067	0.066	0.069
	µg (Na^+^) cm^−2^	0.396	0.411	0.388	0.437
	*ρ*, g cm^−3^	3.809	3.800	3.792	3.807
2	Experimental parameters: 0.50 mol dm^−3^ HNO_3_, 40 °C, 240 h				
	µg (Al^3+^) cm^−2^	0.763	0.738	0.658	0.829
	µg (Ca^2+^) cm^−2^	0.731	0.731	0.667	0.802
	µg (Fe^3+^) cm^−2^	0.015	0.015	0.014	0.016
	µg (Mg^2+^) cm^−2^	0.100	0.099	0.095	0.103
	µg (Na^+^) cm^−2^	0.892	0.878	0.795	0.974
	*ρ*, g cm^−3^	3.776	3.776	3.769	3.783
3	Experimental parameters: 1.25 mol dm^−3^ HNO_3_, 25 °C, 240 h				
	µg (Al^3+^) cm^−2^	1.918	1.833	1.636	2.059
	µg (Ca^2+^) cm^−2^	0.990	1.009	0.932	1.094
	µg (Fe^3+^) cm^−2^	0.029	0.029	0.026	0.032
	µg (Mg^2+^) cm^−2^	0.253	0.255	0.239	0.273
	µg (Na^+^) cm^−2^	1.440	1.440	1.262	1.667
	*ρ*, g cm^−3^	3.783	3.790	3.783	3.798
4	Experimental parameters: 1.25 mol dm^−3^ HNO_3_, 55 °C, 24 h				
	µg (Al^3+^) cm^−2^	1.353	1.294	1.154	1.452
	µg (Ca^2+^) cm^−2^	0.582	0.594	0.548	0.643
	µg (Fe^3+^) cm^−2^	0.019	0.019	0.017	0.020
	µg (Mg^2+^) cm^−2^	0.195	0.197	0.186	0.209
	µg (Na^+^) cm^−2^	1.014	1.016	0.907	1.146
	*ρ*, g cm^−3^	3.784	3.781	3.773	3.788
5	Experimental parameters: 2.00 mol dm^−3^ HNO_3_, 40 °C, 24 h				
	µg(Al^3+^) cm^−2^	1.691	1.636	1.459	1.836
	µg (Ca^2+^) cm^−2^	1.688	1.689	1.540	1.853
	µg (Fe^3+^) cm^−2^	0.025	0.025	0.023	0.027
	µg (Mg^2+^) cm^−2^	0.663	0.653	0.590	0.728
	µg (Na^+^) cm^−2^	1.415	1.451	1.279	1.661
	*ρ*, g cm^−3^	3.778	3.776	3.768	3.783

**Table 10 materials-15-08823-t010:** Verification of experimental and calculated values of eluted ion amounts and alumina ceramic density at assessed optimal corrosion parameters.

No. of Verifications	Response	Experimental Values	Mean Predicted Values	Low CI (95%)	High CI (95%)
1	Experimental parameters: 0.50 mol dm^−3^ HNO_3_, 25 °C, 24 h, desirability 96%				
	µg (Al^3+^) cm^−2^	0.157	0.174	0.150	0.202
	µg (Ca^2+^) cm^−2^	0.278	0.167	0.149	0.186
	µg (Fe^3+^) cm^−2^	0.003	0.004	0.004	0.005
	µg (Mg^2+^) cm^−2^	0.021	0.033	0.032	0.034
	µg (Na^+^) cm^−2^	0.108	0.198	0.186	0.210
	*ρ*, g cm^−3^	3.815	3.805	3.796	3.813
2	Experimental parameters: 0.50 mol dm^−3^ HNO_3_, 40 °C, 24 h, desirability 88%				
	µg (Al^3+^) cm^−2^	0.310	0.321	0.286	0.360
	µg (Ca^2+^) cm^−2^	0.101	0.101	0.092	0.111
	µg (Fe^3+^) cm^−2^	0.006	0.006	0.005	0.006
	µg (Mg^2+^) cm^−2^	0.038	0.038	0.037	0.039
	µg (Na^+^) cm^−2^	0.250	0.247	0.235	0.261
	*ρ*, g cm^−3^	3.775	3.786	3.778	3.793

## Data Availability

Not applicable.

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
