# Peer review of "Statistical Optimisation of Chemical Stability of Hybrid Microwave-Sintered Alumina Ceramics in Nitric Acid"

_materials, 2022, doi:10.3390/ma15248823_

Round 1

Reviewer 1 Report

The authors studied the stability of ceramic metals in corrosive media, and they have explored various properties of corrosive media. in their study, they investigated the chemical stability of the alumina ceramic in various aqueous  HNO3 solutions in addition they employed the Box-Behnken approach to obtain the optimum chemical stability conditions. Therefore, their findings showed a negligible change in the chemical stability of the alumina ceramics. 

Author Response

Response to Reviewer’s Comments

Manuscript reference number: materials-2046299

The paper titled “Statistical optimisation of chemical stability of hybrid microwave-sintered alumina ceramics in nitric acid”

Thank you for the review. We have done our best to correct the article and fulfill all the requirements and questions this time. We are resubmitting the corrected manuscript as suggested by the reviewers. The Track Changes option is used.

Our responses are as follows:

Reviewer #1:

The authors studied the stability of ceramic metals in corrosive media, and they have explored various properties of corrosive media. in their study, they investigated the chemical stability of the alumina ceramic in various aqueous HNO3 solutions in addition they employed the Box-Behnken approach to obtain the optimum chemical stability conditions. Therefore, their findings showed a negligible change in the chemical stability of the alumina ceramics.

Response - Thank you very much for your observation. English language was corrected by Božena Tokić, English language professor at University of Zagreb.

Reviewer 2 Report

This is an interesting paper containing important results which is recommended to publish. Authors are invited nevertheless to amend the manuscript accounting for the following:

1.     The manuscript does not explain how authors have obtained the surface concentration of ions (in μg cm-2) presented in Table 5 and elsewhere. They have identified the content of ions in water solutions contacting samples (lines 151-153) however there are not any explanations or equations in the manuscript to explicit show them and their use.

2.     Were all samples engraved (lines 107-108)? The text “Engraved and sintered alumina pellets were…” (line 141) suggests to readers that there were two sets of samples: engraved and non-engraved.

3.     Lines 161-162: Since the solutions contacting samples were always water solutions the text “…properties of media (acid, basic, static, flowing etc.)…” is better describing the situation by replacing “acid, basic” by “pH” as the solutions can also be neutral.

4.     Line 195: Misprint, this is Table 4 not Table 3.

5.     Table 5: It is needed to give explanation of the meaning of surface concentrations of ions and explain how these were derived.  

Author Response

Response to Reviewer’s Comments

Manuscript reference number: materials-2046299

The paper titled “Statistical optimisation of chemical stability of hybrid microwave-sintered alumina ceramics in nitric acid”

Thank you for the review. We have done our best to correct the article and fulfill all the requirements and questions this time. We are resubmitting the corrected manuscript as suggested by the reviewers. The Track Changes option is used.

Our responses are as follows:

As requested by the Reviewer 2, English language was corrected and improved by Božena Tokić, English language professor at University of Zagreb.

Reviewer #2:

This is an interesting paper containing important results which is recommended to publish. Authors are invited nevertheless to amend the manuscript accounting for the following:

  1. The manuscript does not explain how authors have obtained the surface concentration of ions (in μg cm-2) presented in Table 5 and elsewhere. They have identified the content of ions in water solutions contacting samples (lines 151-153) however there are not any explanations or equations in the manuscript to explicit show them and their use.

Response - Thank you very much for your observation. We have added explanation in sections 2.1. Preparation of Al2O3 ceramics and 2.3. Corrosion monitoring of alumina in aqueous HNO3 solution: “After sintering, the dimensions (diameter and height) of all alumina samples were measured and the alumina area was calculated for each sample” and “The obtained results are expressed as the amount of eluted ions (Mn+: Al3+, Ca2+, Fe3+, Mg2+, and Na+) in µg per square centimetre of the tested alumina area (µg Mn+ cm–2)”.

  1. Were all samples engraved (lines 107-108)? The text “Engraved and sintered alumina pellets were…” (line 141) suggests to readers that there were two sets of samples: engraved and non-engraved.

Response - Thank you very much for your comment. Green cylindrical pellets were engraved with numbers to easily follow each one throughout the experiment. There was one set of samples – engraved. In order to avoid misunderstanding we have deleted “engraved” in manuscript. We corrected it in sections 2.1. Preparation of Al2O3 ceramics and 2.3. Corrosion monitoring of alumina in aqueous HNO3 solution.

  1. Lines 161-162: Since the solutions contacting samples were always water solutions the text “…properties of media (acid, basic, static, flowing etc.)…” is better describing the situation by replacing “acid, basic” by “pH” as the solutions can also be neutral.

Response - Thank you for your remark; it is corrected.

  1. Line 195: Misprint, this is Table 4 not Table 3.

Response - Thank you for your remark; it is corrected.

  1. Table 5: It is needed to give explanation of the meaning of surface concentrations of ions and explain how these were derived.

Response - Thank you very much for your comment. We have added explanation: “The obtained results are expressed as the amount of eluted ions (Mn+: Al3+, Ca2+, Fe3+, Mg2+, and Na+) in µg per square centimetre of the tested alumina area (µg Mn+ cm–2)”.

Reviewer 3 Report

please address all comments in the attached file.

Author Response

Response to Reviewer’s Comments

Manuscript reference number: materials-2046299

The paper titled “Statistical optimisation of chemical stability of hybrid microwave-sintered alumina ceramics in nitric acid”

Thank you for the review. We have done our best to correct the article and fulfill all the requirements and questions this time. We are resubmitting the corrected manuscript as suggested by the reviewers. The Track Changes option is used.

Our responses are as follows:

Reviewer #3:

  1. Please read whole text again and check the grammar and type-writing mistakes.

Response - Thank you for your comment. The manuscript was corrected by colleague Božena Tokić, an English professor at University of Zagreb.

  1. Why did authors choose these concentrations and temperatures?

Response – Thank you for your question. We have added explanation in section 1. Introduction: “It is important to point out that no standardised test procedures are known for the determination of liquid phase corrosion [Breviary Technical Ceramics, Fahner, Hans Verlag, 2004].”and “According to [Breviary Technical Ceramics, Fahner, Hans Verlag, 2004] liquid phase corrosion methods of ceramics include following conditions (i) the storage of test sam-ples in the corrosive media at room temperature (over a test period of between 10 and 100 days), (ii) storage of the test samples in the reflux cooler at higher temperatures (over a test period of between 5 and 50 days) and (iii) storage in autoclaves at higher temperatures and pressure (test period 1 to 5 days).”

  1. Please complete all captions. These are imperfect.

Response – According to Reviewer 6, we have removed Figure 1 from the manuscript.

  1. Please identify the type of corrosion reactions in the ceramics. For example, chemical or electrochemical reactions and other issues.

Response - Thank you for your suggestion. Electrochemical reactions in corrosion process are typical for conductive materials (such as metals and alloys). As we pointed out in manuscript (Section 1. Introduction), corrosion reactions in ceramics are always chemical reactions, more precisely dissolution of ceramics (especially impurities in ceramics)

  1. SiN3 or Si3N4?

Response - Thank you for your observation. We corrected SiN3 to silicon nitride materials in manuscript.

  1. Why? More surface can lead to the more contact of acid?

Response - Thank you for your question. Yes, larger specific surface area leads to more contact with the acidic medium and thus higher corrosion. We added an explanation in the manuscript.

  1. Please make more details about the application of alumina in the nitric acid and the novelty of your work.

Response - Thank you for your comment. The application of alumina in the nitric acid is implied in various industrial processes where ceramic parts are exposed to corrosive media such as aqueous acid solutions. The novelty of our work is detailed optimization of the corrosion process of alumina ceramics in nitric acid as well as the impact of the hybrid microwave sintering technique which has not been investigated in broader sense until now.

  1. What technique did you use to determine the composition? XRF?

Response - Thank you for your question. We have added in manuscript in section 2.1. Preparation of Al2O3 ceramics “Chemical composition was determined by inductively coupled plasma (ICP) spectrometry according to the manufacturer's declaration; it is shown in Table 1.”

  1. It needs to have an alumina sample made by other methods or compare your data with literature.

Response -  Thank you for your advice. We have added in the manuscript comparison of average grain size of same alumina ceramics sintered in electrical and hybrid microwave oven: ” The calculated average grain size of 2.1 µm, determined by the line intercept method [42,43], is in accordance with literature findings [44–47]. The average grain size of the same alumina ceramics sintered in a high-temperature electric kiln (conventional sintering) at the same temperature was greater, i.e., 7.6 μm [24]. The grain size decreases in hybrid microwave sintering compared to conventional sintering due to a shorter sintering time.”

  1. Did you coat your samples before SEM? If yes, what is that? Gold? Please make it clear.

Response - Thank you  for your question. Yes, samples were coated with a layer of gold prior to analysis of morphology. We added it in manuscript.

  1. What about impurity?

Response - Thank you for your question. Corrosion of ceramic materials is influenced by several parameters, one of which is chemical composition (listed in Table 1) as stated in manuscript. In framework of this research, we did not analyze influence of different concentration of impurities. We can include it in future investigation. We provided in manuscript amount of Al2O3 as well as impurities (CaO, Fe2O3, Na2O, SiO2) and sintering aid (MgO).

  1. What is JCPDS numbers?

Response - Thank you for your question. We have added  in section 3.1. “Characteristic peaks in the X-ray diffractogram (Figure 2) show that alumina granules consist of only one crystalline phase, that is α-Al2O3 (corundum) (ICDD PDF#46-1212).”

  1. Please make more material type discussion.

Response - Thank you for your comment. We have added discussion in manuscript.

  1. Is it from a reference or drawn by authors? If it is from a reference, please identify.

Response - Thank you for your comment. Figure 6 was drawn by authors, we improved quality of Figure 6 in manuscript.

  1. Is it an obvious conclusion? Please make some scientific discussions.

Response - Thank you for your comment. We have added in manuscript.

  1. Please wide the table to avoid this type of writing appearance.

Response - Thank you for your suggestion. We corrected the table in manuscript as requested.

Reviewer 4 Report

In this paper, the authors studied the leaching behavior of alumina ceramics sintered with hybrid microwave technique. The authors performed experiments in different time, temperature, and acid solution concentrations. The authors also derived the optimal conditions where the testing specimens can obtain the highest chemical stability. The authors concluded that all three parameters, including time, temperature, and acid concentration play a role in determining the leaching behavior of alumina ceramics. It would be better if in the future work the testing can be conducted at even higher temperatures and longer durations to present more obvious trends. There are some questions and concerns needed to be addressed before the paper can be accepted.

Major concerns:

1, Line 123 and line 179, the XRD pattern of the starting granules is obtained, but the XRD pattern of the sintered specimen is more important and is thus also necessary.

2, Figure 3, the quality of the SEM images is low, and the images are blurry. Improvement in image qualities is recommended.

3, Line 189-191, the SEM images of the indentation in hardness testing is recommended to be included. The formula to calculate hardness is also recommended to be added.

4, Some SEM images and some analysis/discussion of the post-leaching specimens is recommended to be added to section 3.2.

5, Any particular reason that second-order terms are used in the regression equations? From the plots it looks like some surfaces are quite flat, suggesting the relation can be linear. A more complicated regression equation may lead to overfitting of the data.

6, Line 250, This claim may not be obvious because the trend is not obvious in several plots. The vertical axis (amounts of eluted ions) is unnecessarily large so the trend is not easy to observe. The authors are recommended to reduce the range of ion axis.

7, Table 9, is the predicted values calculated from regression equations? If so, it may not be very critical to have a good match of the experimental values with predicted values. It would be more meaningful if a good prediction can be obtained on the unseen data as well.

Minor concerns:

1, Line 86, more introduction is needed for RSM, as well as the Box-Behnken design

2, Line 92, more introduction is needed for microwave sintering as well, some literatures about this technique and its advantages over conventional ones.

3, Line 105, what is the mesh and purity of the starting powder?

4, Line 111 and 112 seems duplicate with line 115-117.

5, Line 132, for Vickers hardness testing, what is the loading period?

6, Section 2.3, for the leaching of alumina, any leaching standard followed?

7, Line 171-173, this is a little confusing. "Six experimental responses were recorded:..., listed in Table 3". However, Table 3 is not about the six responses.

8, Caption of table 3 should be more explanatory.

9, Figure 2, The JCPDS # used to characterize is recommended to be reported and is recommended to be plotted in the Figure 2.

10, Table 3, "0,028" -> "0.028"

11, Some comparisons of the hardness and fracture toughness with literature data is recommended to be added.

12, Table 5, How is the leaching data compared to literature? Does microwave-sintered samples have a better corrosion resistance?

13, The order in Table 3 and Table 5 is recommended to match so it's easier to read.

14, Line 206, some description of ANOVA is recommended to be added to Method section.

15, Figure 5, The authors are recommended to unify the orientation of axes in the plot of each ion. Also, the colormap is missing.

16, Line 242, some description about multicriteria methodology, Derringer function or desirability function is is recommended to be added.

17, Line 251-252, "The increase of HNO3 concentration influenced the increase of eluted amounts of all ions", what influence it is referring to? This is recommended to be plotted too to show the influence.

18, Line 273, "highest corrosion resistance of present alumina at the beginning of the corrosion test", the authors are recommended to add some explanations about why it is.

19, Line 316, "good chemical stability", how to define "good" here? Is it comparing to the conventional sintered samples?

Author Response

Response to Reviewer’s Comments

Manuscript reference number: materials-2046299

The paper titled “Statistical optimisation of chemical stability of hybrid microwave-sintered alumina ceramics in nitric acid”

Thank you for the review. We have done our best to correct the article and fulfill all the requirements and questions this time. We are resubmitting the corrected manuscript as suggested by the reviewers. The Track Changes option is used.

Our responses are as follows:

Reviewer #4:

Major concerns:

  1. Line 123 and line 179, the XRD pattern of the starting granules is obtained, but the XRD pattern of the sintered specimen is more important and is thus also necessary. 1. The Abstract doesn't clearly state the literature gap found, as well as the main motivation to develop this work. Thus, please clearly state the gap found in the literature in the Abstract, Introduction and Conclusions. The main goals are also not clear in the Abstract.

Response - Thank you very much for your comment. We have corrected manuscript. From previous results of XRD analysis of pure alumina ceramics we confirmed that phase composition did not change after sintering by comparison of XRD patterns of alumina raw powder and sintered alumina samples (doi.org/10.3390/su142114244).

  1. Figure 3, the quality of the SEM images is low, and the images are blurry. Improvement in image qualities is recommended.

Response - Thank you very much for your comment. Unfortunately, that is the best image quality we could obtain by used SEM equipment.

  1. Line 189-191, the SEM images of the indentation in hardness testing is recommended to be included. The formula to calculate hardness is also recommended to be added.

Response - Thank you very much for your comment. We have added formula for calculation of the Vickers hardness in the manuscript. The resolution of SEM image of the indentation is low, therefore we did not add in manuscript.

  1. Some SEM images and some analysis/discussion of the post-leaching specimens is recommended to be added to section 3.2.

Response - Thank you very much for your comment. Because of very low dissolution of ceramics in investigated corrosive media, there are no visible changes in surface of post-leaching specimens.

  1. Any particular reason that second-order terms are used in the regression equations? From the plots it looks like some surfaces are quite flat, suggesting the relation can be linear. A more complicated regression equation may lead to overfitting of the data.

Response - Thank you for the observation. To develop the adequate model it is necessary to take into account all relevant model characteristics including residual analysis. So, we obtained the model based on good residual characteristics and the best possible prediction ability.  There was a concern to not overfit the model, but regarding the adjusted R-square we obtained the optimal one. We are not sure if this explanation is excessive to include into the manuscript, but we can include it if you think it will be beneficial.

  1. Line 250, This claim may not be obvious because the trend is not obvious in several plots. The vertical axis (amounts of eluted ions) is unnecessarily large so the trend is not easy to observe. The authors are recommended to reduce the range of ion axis.

Response - Thank you very much for your comment. We enhanced the Figures according to the comment, so the trend and variation is more visible. The Figure 5 is updated. 

  1. Table 9, is the predicted values calculated from regression equations? If so, it may not be very critical to have a good match of the experimental values with predicted values. It would be more meaningful if a good prediction can be obtained on the unseen data as well.

Response - Thank you very much for your comment. The predicted values are the mean predicted values calculated from models for the given set of conditions (parameter values). The confidence intervals of means were also given to fulfill the information. As the verification points are very close to the predicted values, we can accept that the models are valuable. In order to be more precise, we changed the column name “predicted values” to “mean predicted value”.

Minor concerns:

  1. Line 86, more introduction is needed for RSM, as well as the Box-Behnken design.

Response - Thank you very much for your comment. We added description in the manuscript.

  1. Line 92, more introduction is needed for microwave sintering as well, some literatures about this technique and its advantages over conventional ones.

Response - Thank you very much for your comment. We have added in manuscript.

  1. Line 105, what is the mesh and purity of the starting powder?

Response - Thank you very much for your comment. We provided in manuscript the mesh  and purity of the starting powder.

  1. Line 111 and 112 seems duplicate with line 115-117.

Response - Thank you very much for your observation. We have deleted Lines 111 and 112 and left 115 and 117.

  1. Line 132, for Vickers hardness testing, what is the loading period?

Response - Thank you very much for your comment. We have added in section 2.2. Characterization of alumina ceramics: “Vickers hardness measurements (HV1) were performed using indentation loads of 9.807 N for 15 s according to the standard specification ASTM E92-72.”

  1. Section 2.3, for the leaching of alumina, any leaching standard followed?

Response - Thank you very much for your comment. The used ICP – AES, Teledyne Leeman Labs (Hudson, NH, SAD) was calibrated prior to analysis of the concentration of eluted ions (Al3+, Ca2+, Fe3+, Mg2+ and Na+), that is normal procedure.

  1. Line 171-173, this is a little confusing. "Six experimental responses were recorded:..., listed in Table 3". However, Table 3 is not about the six responses.

Response - Thank you very much for your comment. We found that it was a mistake during formatting, so we rephrased the sentence to: “Therefore, six experimental responses were recorded: amounts of Al3+, Mg2+, Ca2+, Na+ and Fe3+ ions eluted from alumina and the density of alumina.”

  1. Caption of table 3 should be more explanatory.

Response - Thank you very much for your comment. The caption was changed to: “Experimental plan according to the Box-Behnken design”.

  1. Figure 2, The JCPDS # used to characterize is recommended to be reported and is recommended to be plotted in the Figure 2.

Response - Thank you very much for your comment. We have added  in section 3.1. “Characteristic peaks in the X-ray diffractogram (Figure 2) show that alumina granules consist of only one crystalline phase, that is α-Al2O3 (corundum) (ICDD PDF#46-1212).”

  1. Table 3, "0,028" -> "0.028"

Response - Thank you very much for your observation. We have corrected it in the manuscript.

  1. Some comparisons of the hardness and fracture toughness with literature data is recommended to be added.

Response - Thank you very much for your comment. We added comparison in the manuscript in section 3.1. Properties of alumina ceramics: “In comparison to the hardness and indentation fracture toughness of the same alumina ceramics sintered in an electric kiln, which amount to 1762 ± 77 and 5.44 ± 0.93 MPa m1/2 (Casellas), respectively, the values for the ceramic samples sintered in a hybrid microwave oven are higher (Table 4) [24].

  1. Table 5, How is the leaching data compared to literature? Does microwave-sintered samples have a better corrosion resistance?

Response - Thank you very much for your comment. We have added in manuscript.

  1. The order in Table 3 and Table 5 is recommended to match so it's easier to read.

Response - Thank you very much for your comment. We matched the order in Tables 3 and 5 and corrected it in manuscript.

  1. Line 206, some description of ANOVA is recommended to be added to Method section.

Response - Thank you very much for your comment. We added description in the manuscript in section 3.2. Modelling of the amount of eluted ions and alumina density the following:

In the case that the p-value is less than 0.05 the model may be considered as significant. In such case, the model succeeded to explain more variability than it remained unexplained.

A polynomial regression model was used to analyze the relationship between dependent and independent variables.

  1. Figure 5, The authors are recommended to unify the orientation of axes in the plot of each ion. Also, the colormap is missing.

Response - Thank you very much for your comment. We enhanced the figures according to the comment, so the trend and variation is more visible, and the orientation of axes is unified. The Figure 5 is updated. 

  1. Line 242, some description about multicriteria methodology, Derringer function or desirability function is recommended to be added.

Response - Thank you very much for your comment. We added description in the manuscript in section 3.3. Optimiszation and verification of alumina ceramics corrosion resistance in nitric acid:

The input data of each individual response (yi) were transformed into an individual dimensionless desirability function, di (desirability scale), ranging from zero (non-desirable) to 1 (highly desired responses), after which any improvement would be insignificant. Taking into account the desirability functions of individual responses (di), it is possible to obtain the overall desirability (D) according to the equation below:

(6)

where D is the overall desirability, dm is the individual desirability function of the m response, and m is the number of responses [30].

  1. Line 251-252, "The increase of HNO3 concentration influenced the increase of eluted amounts of all ions", what influence it is referring to? This is recommended to be plotted too to show the influence.

Response - Thank you very much for your comment. We already provided that data in Table 5.

  1. Line 273, "highest corrosion resistance of present alumina at the beginning of the corrosion test", the authors are recommended to add some explanations about why it is.

Response - Thank you very much for your comment. We have added in manuscript.

  1. Line 316, "good chemical stability", how to define "good" here? Is it comparing to the conventional sintered samples?

Response - Thank you very much for your comment. Very low amount eluted of Al3+, Mg2+, Ca2+, Na+ and Fe3+ ions after the corrosion test of alumina ceramics showed a very good corrosion resistance in the HNO3 aqueous solution. For all investigated conditions of corrosion alumina ceramic samples in aqueous HNO3, the amount of eluted ions in a descending order is as follows: Al > Na > Ca > Mg > Fe.

Reviewer 5 Report

The paper contains interesting information about the chemical resistance of alumina ceramics. The introduction is written in a proper way. The authors did a literature review and highlighted their research gap and novelty. Below I put all (mostly minor) remarks:

1. Line 14. What do you mean by the composition of ceramics? You meant chemical composition???

2. Please put some quantified values of the most important outcomes from your research.

3. In Line 160 please put citations related to mentioned findings.

4. Table 3 - replace the comma with a dot in density value. 

5. Please provide the standard related to your laboratory tests (where it is applicable). 

6. What determined 10 days test regime? Please justify it. 

7. Please improve the quality of Figure 6.

8. Line 316: is "good chemical stability" quantified? Please rewrite the whole conclusion part and put the most important outcomes of your work covered by quantified values. 

Author Response

Response to Reviewer’s Comments

Manuscript reference number: materials-2046299

The paper titled “Statistical optimisation of chemical stability of hybrid microwave-sintered alumina ceramics in nitric acid”

Thank you for the review. We have done our best to correct the article and fulfill all the requirements and questions this time. We are resubmitting the corrected manuscript as suggested by the reviewers. The Track Changes option is used.

Our responses are as follows:

Reviewer #5:

The paper contains interesting information about the chemical resistance of alumina ceramics. The introduction is written in a proper way. The authors did a literature review and highlighted their research gap and novelty. Below I put all (mostly minor) remarks:

  1. Line 14. What do you mean by the composition of ceramics? You meant chemical composition???

Response - Thank you for your remark. We have added in the manuscript ” chemical and phase composition of the ceramics”.

  1. Please put some quantified values of the most important outcomes from your research.

Response – Thank you very much for your comment. We have added in manuscript quantified values of the most important outcomes from our research.

  1. In Line 160 please put citations related to mentioned findings.

Response - Thank you very much for your comment. We have corrected in manuscript.

  1. Table 3 - replace the comma with a dot in density value.

Response - Thank you for your remark; it is corrected.

  1. Please provide the standard related to your laboratory tests (where it is applicable).

Response - Thank you very much for your suggestion. We have added it in the manuscript.

  1. What determined 10 days test regime? Please justify it.

Response - Thank you very much for your observation. There is no standards procedure for testing corrosion properties of ceramics materials in nitric acid. We have added in section 1. Introduction: “. Methods for determining the liquid-phase corrosion of ceramics include the following conditions: (i) the storage of test samples in the corrosive media at room temperature (over a test period of between 10 and 100 days); (ii) storage of the test samples in the reflux cooler at higher temperatures (over a test period of between 5 and 50 days); and (iii) storage in autoclaves at higher temperatures and pressure (test period 1 to 5 days) [15].”

  1. Please improve the quality of Figure 6.

Response - Thank you for your remark. We have improved the quality of Figure 6.

  1. Line 316: is "good chemical stability" quantified? Please rewrite the whole conclusion part and put the most important outcomes of your work covered by quantified values.

Response – Thank you very much for your comment. We have rewritten the whole conclusion part and put the most important outcomes of our work covered by quantified values.

Reviewer 6 Report

Dear Authors,

Congratulations on your work, which is focused on a very interesting subject. As any other paper in this phase, there are some amendments to do, whose can improve the overall quality of your paper. Thus, I'm providing below some comments and suggestions, trying to collaborate by this way in improving your paper:

1. The Abstract doesn't clearly state the literature gap found, as well as the main motivation to develop this work. Thus, please clearly state the gap found in the literature in the Abstract, Introduction and Conclusions. The main goals are also not clear in the Abstract.

2. There is a lack of quantitative results in the Abstract. Please improve.

3. The novelty brought by your work is also not properly pointed out. Thus, please state clearly the novelty that your paper represents for the scientific community, stating as well if your contribution is exclusively scientific or if there was some practical motivation behind the development of your work. Any industrial application based on this work should also be pointed out.

3. Figure 1 doesn't bring value to the paper. Please remove it.

4. The Literature Review is well done, but readers prefer direct speech, describing briefly in what the work of previous Researchers has been focusedon, methodology used and main results. Please avoid as much as possible generic ideas.

5. Throughout the Methods, no standards are pointed out to guide your characterization. Could you point out the standards followed in your work?

6. I'm not sure if Figure 2 should remain in section 3.1 or if it must be transferred to chapter 2, because this characterizes the raw material. Moreover, and considering that Figure 2 remains in 3.1, the XRD characterization is not the first analysis to take care. Usually, morphology is firstly analysed.

7. It is not clear as the values of the Al2O3 mechanical characterization appear in Table 3. Please explain (mainly the fracture toughness value). 

8. The DISCUSSION is really poor. In fact, only references [8,13,20,44] are used to compare and explain some of the phenomena registered. Please enlarge your discussion, bring other similar works and corresponding results to the discussion.

9. Please highlight your main achievements using bullet points in the Conclusions.

Best wishes.

Kkind regards,

FGS

Author Response

Response to Reviewer’s Comments

Manuscript reference number: materials-2046299

The paper titled “Statistical optimisation of chemical stability of hybrid microwave-sintered alumina ceramics in nitric acid”

Thank you for the review. We have done our best to correct the article and fulfill all the requirements and questions this time. We are resubmitting the corrected manuscript as suggested by the reviewers. The Track Changes option is used.

Our responses are as follows:

Reviewer #6:

Congratulations on your work, which is focused on a very interesting subject. As any other paper in this phase, there are some amendments to do, whose can improve the overall quality of your paper. Thus, I'm providing below some comments and suggestions, trying to collaborate by this way in improving your paper:

  1. The Abstract doesn't clearly state the literature gap found, as well as the main motivation to develop this work. Thus, please clearly state the gap found in the literature in the Abstract, Introduction and Conclusions. The main goals are also not clear in the Abstract.

Response - Thank you very much for your comment. We have added in the manuscript.

  1. There is a lack of quantitative results in the Abstract. Please improve.

Response - Thank you very much for your comment. We have added in the Abstract quantified values of the most important outcomes from our research.

  1. The novelty brought by your work is also not properly pointed out. Thus, please state clearly the novelty that your paper represents for the scientific community, stating as well if your contribution is exclusively scientific or if there was some practical motivation behind the development of your work. Any industrial application based on this work should also be pointed out.

Response - Thank you very much for your comment. We have added in the manuscript.

  1. Figure 1 doesn't bring value to the paper. Please remove it.

Response - Thank you for your remark. We have removed Figure 1.

  1. The Literature Review is well done, but readers prefer direct speech, describing briefly in what the work of previous Researchers has been focused on, methodology used and main results. Please avoid as much as possible generic ideas.

Response - Thank you very much for your comment. We have corrected it.

  1. Throughout the Methods, no standards are pointed out to guide your characterization. Could you point out the standards followed in your work?

Response - Thank you very much for your comment. We have added in manuscript.

  1. I'm not sure if Figure 2 should remain in section 3.1 or if it must be transferred to chapter 2, because this characterizes the raw material. Moreover, and considering that Figure 2 remains in 3.1, the XRD characterization is not the first analysis to take care. Usually, morphology is firstly analysed.

Response - Thank you very much for your observation. We have left Figure 2 in section 3.1. because we have performed XRD analysis to confirm phase composition of investigated alumina ceramic, after that we have performed morphological characterization.

  1. It is not clear as the values of the Al2O3 mechanical characterization appear in Table 3. Please explain (mainly the fracture toughness value).

Response - Thank you very much for your observation. We have corrected it (mechanical properties are presented in Table 4).

  1. The DISCUSSION is really poor. In fact, only references [8,13,20,44] are used to compare and explain some of the phenomena registered. Please enlarge your discussion, bring other similar works and corresponding results to the discussion.

Response - Thank you very much for your comment. We have enlarged our discussion.

  1. Please highlight your main achievements using bullet points in the Conclusions.

Response - Thank you very much for your comment. We have corrected conclusion.

Round 2

Reviewer 3 Report

Authors addressed the comments. Then, the manuscript can be published.

Reviewer 4 Report

After the revision, the quality of the paper has been significantly improved. A lot of details have been added to enhance the soundness and readability of the paper.